# Targeted c-Myc Inhibition and Systemic Temozolomide Therapy Extend Survival in Glioblastoma Xenografts

**DOI:** 10.3390/bioengineering10060718

**Published:** 2023-06-14

**Authors:** Laxmi Dhungel, Cayla Harris, Lauren Romine, Jan Sarkaria, Drazen Raucher

**Affiliations:** 1Department of Cell and Molecular Biology, University of Mississippi Medical Center, 2500 North State Street, Jackson, MS 39216, USA; ldhungel@umc.edu (L.D.); charris1@umc.edu (C.H.); lkennedy522@gmail.com (L.R.); 2Division of Radiation Oncology, Mayo Clinic and Foundation, 200 First Street, SW, Rochester, MN 55905, USA; sarkaria.jann@mayo.edu

**Keywords:** glioblastoma, targeted therapy, combination therapy, polypeptide

## Abstract

Glioblastoma is a highly aggressive disease with poor patient outcomes despite current treatment options, which consist of surgery, radiation, and chemotherapy. However, these strategies present challenges such as resistance development, damage to healthy tissue, and complications due to the blood–brain barrier. There is therefore a critical need for new treatment modalities that can selectively target tumor cells, minimize resistance development, and improve patient survival. Temozolomide is the current standard chemotherapeutic agent for glioblastoma, yet its use is hindered by drug resistance and severe side effects. Combination therapy using multiple drugs acting synergistically to kill cancer cells and with multiple targets can provide increased efficacy at lower drug concentrations and reduce side effects. In our previous work, we designed a therapeutic peptide (Bac-ELP1-H1) targeting the c-myc oncogene and demonstrated its ability to reduce tumor size, delay neurological deficits, and improve survival in a rat glioblastoma model. In this study, we expanded our research to the U87 glioblastoma cell line and investigated the efficacy of Bac-ELP1-H1/hyperthermia treatment, as well as the combination treatment of temozolomide and Bac-ELP1-H1, in suppressing tumor growth and extending survival in athymic mice. Our experiments revealed that the combination treatment of Bac-ELP1-H1 and temozolomide acted synergistically to enhance survival in mice and was more effective in reducing tumor progression than the single components. Additionally, our study demonstrated the effectiveness of hyperthermia in facilitating the accumulation of the Bac-ELP1-H1 protein at the tumor site. Our findings suggest that the combination of targeted c-myc inhibitory biopolymer with systemic temozolomide therapy may represent a promising alternative treatment option for glioblastoma patients.

## 1. Introduction

Glioblastoma is a devastating disease that continues to pose a significant challenge in the field of oncology. Despite significant advances in treatment options over the past few decades, the median survival rate for glioblastoma patients remains low. Surgery, radiation, and chemotherapy are the current standard of care for patients with glioblastoma. Surgery is the primary treatment option, which involves removing bulk and resistant tumors, followed by radiation therapy to eliminate tumor cells that have invaded surrounding areas, and chemotherapy with drugs such as temozolomide to kill any remaining cancer cells [1,2]. However, despite these efforts, the survival rate for glioblastoma patients is still extremely low. The five-year survival rate is just 5%, indicating that there is an urgent need for more effective treatment strategies to improve patient outcomes [3]. One of the major challenges of treating glioblastoma is that it develops resistance to therapy. Another challenge is that current treatments can also damage healthy cells, further compromising the patient’s quality of life. Additionally, the highly migratory nature of glioblastoma and the presence of the blood–brain barrier make treatment more difficult [3,4]. Moreover, glioblastoma is often already well-established by the time the patient develops symptoms and becomes aware of the disease, making preventive strategies challenging [5]. Therefore, there is an urgent need for additional treatment strategies that can be used alone or in combination with existing therapies to selectively target tumor cells, minimize resistance development, and ultimately improve the survival of patients.

Oncogenes are genes that are involved in abnormal and uncontrolled cell proliferation, which can lead to the development of tumors [6]. These oncogenes can arise from mutations or abnormal expressions of their corresponding proto-oncogenes, which are typically involved in normal cell proliferation [6,7]. The MYC family of oncogenes are overexpressed in several types of cancers, including glioblastoma, and suppressing these proteins has been shown to reduce brain tumors [8]. The MYC protein becomes transcriptionally active when it binds to the MYC associated factor X (MAX), and this MYC-MAX complex, also known as the master transcriptional regulator, then binds to enhancer box (E-BOX) sequences that can regulate transcription of a large set of genes related to cell growth and division [8,9,10]. Deregulated expression of MYC proteins can cause uncontrolled cell proliferation and tumor development. In our previous work, our lab designed a therapeutic peptide (Bac-ELP1-H1) that targets the c-myc oncogene and prevents the c-MYC/MAX interaction [4,11,12]. The peptide consists of three components: Bac, ELP, and H1. The Bac segment includes a cell penetrating peptide that facilitates the penetration of the peptide into the cell. The ELP, or elastin-like polypeptide, is thermally responsive and can aggregate at higher temperatures, making it useful for thermal targeting to accumulate peptides at tumor sites. The H1 segment is a c-myc inhibitory peptide that interferes with c-myc/max interaction, resulting in the killing of tumor cells. These studies have shown that this peptide reduces brain tumor sizes, delays neurological deficits related to glioblastoma, and enhances survival in a rat glioblastoma model [4]. Therefore, targeting the MYC oncogene could be a promising approach for treating glioblastoma.

Currently, temozolomide is considered the “gold standard’ chemotherapeutic agent for the treatment of glioblastoma [3,5,13]. However, the use of temozolomide is associated with challenges related to resistance development and severe side effects. To mitigate these challenges, combination therapy that includes the combination of drugs that can act synergistically to kill cancer cells can be used. This combination therapy can have multiple targets and can be used in lower concentrations of each drug, thus providing higher effectiveness and lower side effects compared to the use of a single drug [14,15,16,17]. Our previous study demonstrated the efficacy of Bac-ELP1-H1 and hyperthermia in tumor suppression and survival of rats implanted with rat C6 glioma cells [4]. In this study we expanded our research to the U87 glioblastoma cell line. This cell line is a commonly used experimental model for studying human glioblastoma due to its relevance to human disease [18]. These cells exhibit characteristics that are similar to human glioblastoma cells, including the ability to form tumors when implanted into immunodeficient mice, high levels of proliferation and invasion, and the ability to recapitulate the heterogeneity and molecular complexity of human glioblastoma [19,20]. Moreover, the U87 cell line has been extensively studied and characterized, making it a valuable model for preclinical evaluation of new therapies for glioblastoma. In addition, it is widely available, easy to culture, and can be genetically modified to investigate the role of specific genes or pathways in glioblastoma development and progression. Overall, the U87 xenograft model has become a valuable tool in glioblastoma research, providing a platform for evaluating novel therapeutic agents and investigating the molecular mechanisms underlying glioblastoma pathogenesis [19,20].

Therefore, in this study, we implanted human U87 glioma cells into athymic mice to study the efficacy of Bac-ELP1-H1/hyperthermia and combination treatment of temozolomide and Bac-ELP1-H1 in tumor suppression and survival. Our experiments demonstrated that the combination treatment of Bac-ELP1-H1 and temozolomide acts synergistically to enhance survival in mice and has higher effectiveness in the reduction of tumor progression compared to single treatment with its components. Furthermore, it also demonstrates the effectiveness of hyperthermia in the accumulation of the Bac-ELP1-H1 protein in the tumor sites. This type of targeted therapy can provide better treatments than non-targeted therapy, as it acts on tumor sites and can have minimal effects on normal healthy cells. We believe this approach of combination and targeted therapy using the Bac-ELP1-H1 protein and temozolomide has shown promising results for the treatment of glioblastoma in mice and has potential application for the treatment of human glioblastoma in the future.

In conclusion, the findings of this study suggest that the combination therapy of temozolomide and Bac-ELP1-H1 has the potential to be a more effective treatment option for glioblastoma. However, further research is needed to fully understand the efficacy, safety, and mechanisms of action of this treatment in human patients.

## 2. Materials and Methods

### 2.1. Protein Synthesis and Purification

The Bac-ELP1-H1 polypeptide (MRRIRPRPPRLPRPRPRPLPFPRPGGCYPG-(VPGXG)n-WPGSGNELKRAFAALRDQI) was synthesized and purified using *E. coli* cells that were transformed with the corresponding plasmid [21,22]. Frozen cell stocks were first grown in Circle Grow media with ampicillin (100 ug/mL) at 37 °C for 24 h. The cells were then inoculated into Terrific Broth media with ampicillin (100 ug/mL) and incubated at 37 °C with 220 rpm agitation for protein synthesis. After 16–18 h, the cells were harvested by centrifugation at 6000 RPM for 15 min and resuspended in PBS with beta-mercaptoethanol (BME). The cells were sonicated to release the protein, and bacterial cell impurities were removed by centrifuging at 13,000 RPM for 45 min. A 0.5% *w*/*v* solution of polyethylene imine was added and centrifuged at 11,000 RPM for 30 min to remove any nucleic acid impurities. Salt (NaCl) was added to the supernatant to salt out the protein, which was heated to 42 °C and subsequently centrifuged at 11,000 RPM at room temperature to collect the aggregated protein pellets. An alternate hot and cold spin cycle was used for further purification, utilizing the protein’s nature of forming aggregates at high temperatures and dissolving into solution at low temperatures. The cold spin cycle involved centrifugation at low temperature to remove impurities, while the hot spin cycle involved heating the protein on a water bath (42 °C) to allow protein aggregation and further centrifugation at room temperature to collect the protein pellet. This cycle was repeated three to five times. The final protein concentration and purity were quantified using a spectrometer (UV-1600, Shimadzu, Singapore).

### 2.2. Assessment of the Antiproliferative Effects and Synergistic Interactions of Combination Therapy In Vitro

To determine the antiproliferative effects of the combination treatment of Bac-ELP1-H1 and TMZ on U87 cells, the cell viability of U87 cells on combination treatment was compared to the effect when treated alone with its single components (Bac-ELP1-H1, TMZ). Briefly, U87 (500 cells) were plated on a 96-well plate and treated with different concentrations of Bac-ELP1-H1 (1, 2, 4, 8, 16 μM), TMZ (0, 40, 80, 160 μM), and the combination of these concentrations of Bac-ELP1-H1 and TMZ at 24 h. After 5 d of treatment, the effects of Bac-ELP1-H1, TMZ, and their combination on cell viability were assessed using an MTT ((3-(4,5-dimethylthiazol-2-yl)-2,5-diphenyltetrazolium bromide) assay [23]. The MTT is a colorimetric assay that measures the metabolic activity of live cells. The absorbance of each treatment was normalized to the control (DMSO-treated cells) to determine the percentage survival of cells.

The synergy between temozolomide and Bac-ELP1-H1 for U87 cells based on the proliferation study was determined by using SynergyFinder, an online-based tool that provides synergy scores based on synergy reference models such as HSA, Bliss, Loewe, and ZIP [24]. The highest single agent (HSA) model assumes synergy between drugs if the combination of drugs has higher effects than the maximum effect of its single components [24,25]. The Bliss synergy score assumes synergy if the combination of drugs has higher effects than the multiplicative effects of its single components [24,26]. The Loewe model assumes synergy if the combination of drugs has higher effects than the additive effects of its components. The Bliss synergy score assumes that the single drug acts independently whereas the Loewe synergy score assumes that the single drugs are the same compound [24]. The zero interaction potency model (ZIP) compares the dose-response curves between the combination drugs and their single components and assumes that they do not affect the potency of each other [24,25]. The synergy score obtained for the combination treatment of temozolomide and Bac-ELP1-H1 on U87 cells based on synergy reference models (HSA, Bliss, Loewe, and ZIP) were analyzed and interpreted as synergistic and antagonistic (synergy score < −10—antagonistic; synergy score > 10—synergistic) [24,25].

The results from SynergyFinder were further validated using Compusyn software 1.0 by Chou and Martin [24,27]. The effects for different concentrations of single and the combination of drugs were entered into the software to obtain the combination index (CI) values. The synergy, antagonism, and additive effects were determined based on the combination index (CI) values for the combination of these drugs (synergism (CI < 1), antagonism (CI > 1), additive (CI = 1)).

### 2.3. Animal Experiments

Animal experiments were conducted in compliance with the guidelines of the Institutional Animal Care and Use Committee (UMMC-IACUC) of the Mississippi Medical Center. All procedures that might cause pain or discomfort to animals were performed under strict anesthesia. U87 cells containing green fluorescent protein (GFP) were cultured to confluence in T75 flasks. On the day of surgery, cells were trypsinized and resuspended in PBS at a concentration of 100,000 cells/μL. An athymic nude mouse was injected intraperitoneally with ketamine and allowed sufficient time for anesthesia to take effect. The reflex and status of the animal under anesthesia were observed by performing a firm toe pinch. Tumor implantation was carried out according to the procedure described by Bidwell et al., with some modifications [4]. Povidone iodide was applied to the mouse’s head, and a midline incision was made using a scalpel. A small hole (1.6 mm in diameter) was drilled into the left side of the skull, 1.0 mm forward and 2.0 mm lateral from the bregma. The mouse was then placed on a sterile gauze and fixed to a stereotaxic frame. Tumor cells (U87-GFP; 300,000 cells) were injected 3 mm deep using a Hamilton syringe needle. Bone wax was used to seal the hole, and the wound was closed with 2–3 sutures using 4-0 Vicryl suture. The progression of the tumor was evaluated weekly using an IVIS Spectrum animal imager, as described below.

### 2.4. Evaluation of the Efficacy of Bac-ELP1-H1 and Temozolomide Combination Therapy on Tumor Regression

The athymic nude mice with tumors were treated with peptide Bac-ELP1-H1 and temozolomide and the efficacy of this combination treatment was evaluated by measuring tumor sizes and overall survival. The mice were injected intravenously with peptide Bac-ELP1-H1 for five consecutive days, combined with oral administration of TMZ (20 mg/kg TMZ). The criterion for determining the doses of TMZ (32–327 mg/kg BW) administered via oral gavage and BAC-ELP-H1 (30 mg/kg BW) administered intravenously was based on a combination of factors, including the previously established therapeutic index (CI) and relevant preclinical studies [4,12]. We also conducted a maximum tolerated dose (MTD) study. The MTD study allowed us to determine the highest dose of TMZ (Temozolomide) and BAC-ELP-H1 that could be administered without causing unacceptable toxicity or adverse effects in the experimental animals. By aligning with the established CI, we aimed to ensure an optimal therapeutic dosage range that maximizes the potential efficacy of the treatment while minimizing the likelihood of adverse effects. GBM tumor volumes were examined thrice weekly for 3–6 weeks by quantifying luciferase activity as described below. Animal deaths were recorded twice a day for the entire experimental period. To determine whether treatment with test agents delays the onset of neurological deficits, mice bearing intracerebral GBM tumors were monitored for any development of neurological deficits during the tumor reduction study.

### 2.5. Investigation of the Influence of Hyperthermia on the Biodistribution and Tumor Accumulation of the Bac-ELP1-H1 Protein

The impact of hyperthermia on the biodistribution and tumor accumulation of the Bac-ELP1-H1 protein was assessed by comparing the fluorescence intensities of the rhodamine-labeled protein at different major organs and tumor sites in heated and unheated mice. Intravenous injection of the rhodamine-labeled protein was performed on mice in both the heated and unheated groups. In the heated group, an external heat using Laser Sys-stim (42 °C) (Mettler Electronics Corp, Anaheim, CA, USA) was applied to the tumor site for a total of 2 h, which included cycles of 20 min heating followed by 10 min cooling. After 4 h of protein administration, the mice were sacrificed, and major organs, including the brain, heart, liver, spleen, and kidneys, were harvested. An ex vivo imaging was performed on these organs using IVIS spectrum animal imager, and fluorescence intensities were measured to determine the biodistribution of the protein. The brains were rapidly frozen for further study on tumor accumulation. Cryo-microtome (15 μM) was used to section the frozen brains, which were then mounted on slides and imaged using confocal microscopy. The tumor, which was GFP-labeled, exhibited green fluorescence, whereas the rhodamine-labeled protein exhibited red fluorescence. The distribution of green and red fluorescence on the brain was compared to determine the tumor accumulation of the protein.

### 2.6. In Vivo Assessment of Tumor Growth, Survival, and Therapeutic Efficacy following Implantation of U87 Cells in Mouse

To determine tumor progression, the tumor sizes of the mice were measured twice a week after intraperitoneal injection of D-luciferin. The U87 cells that are implanted into mice expresses luciferase; hence, the reaction of luciferase with luciferin generates bioluminescent light. The intensity of this bioluminescent light corresponds to the number of tumor cells and size. As the mice had GFP-containing tumors, they luminesced upon administration of luciferin. The average radiance, which corresponds to tumor size, was quantitated using the IVIS imaging system. Physical examination for the health and neurological deficits of the mice was also performed. For the survival study, the researchers recorded the number of days that the mice lived for each treatment group and compared the data. Mice that lost more than 20% of their weight were sacrificed and their number of survival days was recorded. The survival data were plotted on GraphPrism software to obtain Kaplan–Meier survival curves [28].

### 2.7. Statistical Analysis

The data obtained from the experiments were analyzed using GraphPrism software 9.5.1 (GraphPad Software, Boston, MA, USA). Statistical analysis was performed using Student’s *t*-test and paired *t*-test for Bac-ELP1-H1 tumor uptake and tumor progression, respectively. Kaplan–Meier survival curves were generated for the survival study, and a log rank test was used to compare the statistically significant differences in survival between treatment groups.

## 3. Results

### 3.1. Bac-ELP1-H1 and TMZ Combination Therapy Exhibits Synergistic Effects on U87 Cells in an In Vitro Model

To assess the impact of the combined Bac-ELP1-H1 and temozolomide treatment on cell proliferation in vitro, we compared the percentage survival of U87 cells following combination treatment to that of cells treated with each compound individually. Our findings reveal that the combination treatment of Bac-ELP1-H1 and TMZ led to a reduced cell viability compared to the cells treated with single components (Figure 1). Furthermore, we employed SynergyFinder and Compusyn software to test the synergy between Bac-ELP1-H1 and TMZ on U87 cells. The SynergyFinder analysis demonstrated that all the tested combination treatments of Bac-ELP1-H1 (2, 4, 8, 16) and TMZ (40, 80, 160) exhibited synergistic effects on U87 cells, as per the Loewe and HSA synergy reference models. Additionally, using the Compusyn software by Chou and Martin, we observed synergy for all the tested combination treatments of Bac-ELP1-H1 and TMZ.

### 3.2. Bac-ELP-H1 Exhibits Localization in Brain Tumors

To determine the localization of Bac-ELP-H1 in brain tumors, rhodamine-labeled Bac-ELP-H1 was intravenously injected into mice bearing intracerebral tumors. Four hours post-injection, FITC-dextran was injected to mark the perfused vasculature. Brains were removed, rapidly frozen, sectioned with a cryomicrotome, and examined using fluorescence microscopy. The injection of high molecular weight FITC-dextran demonstrated an increase in perfused vasculature in the tumor compared to adjacent normal brain tissue. Fluorescence images of brain sections from the experimental group, taken with identical parameters, showed bright staining of tumors relative to autofluorescence, indicating the presence of Bac-ELP-H1 polypeptides within the tumor (Figure 2).

### 3.3. Thermal Targeting of CPP-ELP1-H1 Significantly Increases Tumor Uptake

In our previous work, we demonstrated that Bac-ELP-H1 localizes in rat brain tumors and that its tumor uptake can be enhanced by hyperthermia. To determine whether this targeting method is also effective in a mouse model of glioblastoma, we conducted further experiments. Tumors in mice were heated using the previously established thermal cycling protocol, and tumor deposition was assessed by ex vivo imaging of the mice brains and other organs four hours after injection using an IVIS Spectrum animal imager. Quantitative analysis of the tumors’ fluorescence intensity revealed that thermal targeting significantly increased Bac-ELP-H1-rhodamine tumor accumulation by two-fold (*p* = 0.0001, Student’s *t*-test), as shown in Figure 3. The ex vivo fluorescence analysis of all major organs revealed a noticeable accumulation of Bac-ELP-H1-rhodamine in the kidneys, possibly attributed to the clearance of the polypeptide or its fragments. Similarly, high levels of accumulation were observed in the liver. However, despite these findings, no morphological changes or indications of renal injury were observed in the histopathological examination when compared to the control mice. These results strongly suggest that the proposed treatment is well-tolerated and does not induce renal toxicity, as evidenced by the absence of morphological alterations and the preservation of normal kidney tissue architecture. It is important to note that hyperthermia at the tumor site did not influence the levels of the polypeptide in any other organ. The consistency of our results between the quantitative fluorescence analysis of tissue sections and the ex vivo fluorescence analysis of major organs suggests that this method is sufficient for determining biodistribution. Overall, these findings confirm that Bac-ELP-H1 localizes in brain tumors in a mouse model of glioblastoma and that thermal targeting can enhance tumor uptake of this polypeptide.

### 3.4. Bac-ELP-H1 and Temozolomide Combination Therapy Increases Treatment Efficacy and Survival

To evaluate the effect of Bac-ELP-H1 and TMZ in vivo, U87 MG-luciferase cells were injected into the brains of athymic mice. Once tumors developed, mice were treated with either Bac-ELP-H1, TMZ, a combination of both, or saline as a control. The treatment with Bac-ELP-H1 or TMZ alone resulted in a significant reduction in tumor growth and improved overall survival in comparison to the control group (*p* < 0.05), as shown in Figure 4. The median survival of mice treated with TMZ was 57 days, while that of Bac-ELP-H1 was 53 days, as compared to the control group with a median survival of 45 days. Interestingly, the combination therapy with Bac-ELP-H1 and TMZ demonstrated superior efficacy, significantly inhibiting tumor growth (*p* < 0.005) as compared to the individual treatments. Additionally, the median survival of mice treated with the combination therapy was extended to 70 days, suggesting a potent synergistic effect between Bac-ELP-H1 and TMZ. These findings highlight the therapeutic potential of the combination therapy with Bac-ELP-H1 and TMZ for the treatment of brain tumors.

## 4. Discussion

Glioblastoma is a malignant brain tumor that has a low survival rate despite treatment. Therefore, there is a pressing need to develop novel approaches with higher efficacy in reducing tumor progression and enhancing patient survival. These approaches may involve the discovery of new treatment agents or novel utilization of current treatment strategies. Temozolomide is the current “drug of choice” for the treatment of glioblastoma. However, there are several challenges associated with its use, including the development of drug resistance and side effects. Combination therapy, which involves the use of two or more drugs to treat cancer, may have significant advantages over monotherapy, such as multiple targets, greater ability to kill resistant cells, and fewer side effects. However, the combination of drugs needs to act synergistically to achieve the optimum results.

In this study, we investigated the efficacy of targeted drug delivery of an inhibitory biopolymer (Bac-ELP1-H1) in conjunction with systemic chemotherapy (TMZ) in the suppression of tumor growth and enhancement of survival in a mouse glioblastoma model. We initially tested the synergy between the combination of drugs in vitro in human brain cancer cells (U87) and found that the combination of drugs had higher antiproliferative effects against U87 cells compared to their single components. Furthermore, we used the software SynergyFinder 3.0 and Compusyn 1.0 to test the synergy of the drug combinations, and all the tested drug combinations were found to be synergistic. The combination treatment with Bac-ELP1-H1 and TMZ was then conducted in vivo in athymic nude mice implanted with the U87 tumor. We observed significantly higher efficacy of the combination treatment in reducing tumor growth and enhancing survival compared to treatment with its single components (Bac-ELP1-H1 or TMZ) and the control. This study provides evidence that the combination of targeted drug delivery of an inhibitory biopolymer (Bac-ELP1-H1) with systemic chemotherapy (TMZ) can lead to a synergistic effect that enhances the efficacy of glioblastoma treatment and improves patient survival.

Several studies have investigated the combination treatment of temozolomide with other anticancer drugs, such as lomustine and aldoxorubicin, to improve treatment efficacy in glioblastoma patients [29,30]. Herrlinger et al. [30] conducted an epidemiological study and found that the combination treatment of temozolomide with lomustine had higher overall median survival compared to the patients treated with temozolomide only. However, combination treatment with lomustine-temozolomide exhibited a moderately higher toxicity consisting in symptoms such as brain oedema, speech impairment, sensory dysfunction, and low-grade alopecia compared to treatment with temozolomide only. By contrast, Ros et al. [29] conducted an in vivo study and found no significant difference in reduction of tumor growth and overall survival percentage in mice treated with combination treatment of aldoxorubicin and temozolomide compared to those treated with temozolomide only, although a delay in survival was observed for the combination treatment compared to the single treatment with temozolomide. Similarly, studies have tested the effects of combination treatment of temozolomide with other medications such as penfluridol and metformin, which are antipsychotic and antidiabetic agents, respectively [31,32]. Kim et al. [31] demonstrated that the combination treatment of temozolomide with penfluridol suppressed tumors and enhanced survival in a mouse glioblastoma model. Similarly, Lee et al. [32] showed that the combination treatment of temozolomide with a high dose of metformin significantly enhanced survival in mice with glioblastoma compared to treatment with temozolomide only. However, use of these agents as an anticancer treatment can be associated with serious side effects, and the balance between the merits and side effects associated with the dose needed for cancer treatment needs to be evaluated. Another study by Netto et al. [33] demonstrated the higher effectiveness of combination treatment of temozolomide with plant derivatives such as Matteucinol obtained from *Miconia chamissois* Naudin in tumor reduction in chick embryo chorioallantoic membrane (CAM) compared to untreated groups. However, combination treatment of temozolomide with these plant derivatives requires further investigations regarding bioavailability, mechanism of actions, and ability to cross the blood–brain barrier.

In our previous study, we demonstrated the potential of therapeutic peptides (Bac-ELP1-H1) with hyperthermia for tumor suppression and enhanced survival in a rat glioma model [4]. Our current study built on this work by utilizing a more clinically relevant human U87 glioma cell line implanted into athymic nude mice. Here we have shown that the targeted drug delivery of an inhibitory biopolymer (Bac-ELP1-H1) using hyperthermia can cross the blood–brain barrier and can localize in the brain. The biopolymer is designed to inhibit specific cancer pathways (MYC) and thus has greater specificity to cancer cells and has lesser off-site targets on healthy normal cells. The biodistribution study showed higher uptake of protein on tumor sites when hyperthermia was applied, thereby suggesting higher efficacy of this targeted drug delivery approach in tumor suppression and cancer treatment. Furthermore, the c-myc inhibitor acts synergistically with temozolomide against U87 brain cancer cells, suppresses tumor growth, and increase survival in an orthotopic model of U87 glioblastoma.

The proposed approach utilizes hyperthermia in combination with therapeutic peptides, leveraging the readily available technology of MRI-guided highly focused ultrasound in clinical settings. Hyperthermia has been extensively studied and utilized, demonstrating both safety and feasibility in various applications [34,35]. Additionally, it has shown the ability to sensitize tumor cells to chemotherapy, often employed in combination treatments [36,37,38]. Thus, the technical feasibility of utilizing hyperthermia is well-established. Another crucial aspect to consider is the galenic constraints associated with the therapeutic agent, in this case, ELP. ELP exhibits favorable characteristics, including stability, solubility, bioavailability, compatibility with other components, and non-immunogenicity. These properties make it suitable for the desired route of administration. Consequently, the proposed approach can be readily translated to human studies without encountering any formulation-related challenges or constraints. In summary, the combination of hyperthermia and therapeutic peptides offers a promising avenue for treatment. Hyperthermia has proven safety and feasibility, while ELP addresses the necessary formulation requirements. These factors contribute to the ease of translating the proposed approach to human studies without significant hurdles related to formulation or formulation-related factors.

In conclusion, the combination of targeted drug delivery of the Bac-ELP1-H1 biopolymer and systemic chemotherapy with temozolomide holds great promise in the suppression of tumors and improvement of survival rates in mice. These findings pave the way for the development of a potential combination therapy for the treatment of glioblastoma in humans. The results of this study suggest that this combination therapy can be an effective and safe approach for improving the efficacy of temozolomide in glioblastoma treatment. Thus, the targeted delivery of the Bac-ELP1-H1 biopolymer along with systemic chemotherapy provides a new direction in the development of innovative therapeutic strategies for the treatment of glioblastoma.

## Figures and Tables

**Figure 1 bioengineering-10-00718-f001:**
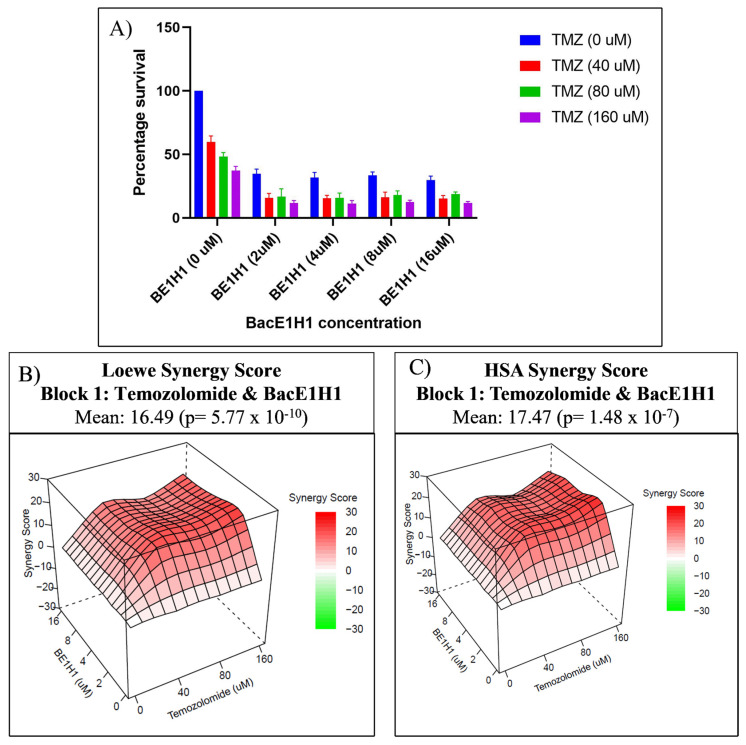
Synergistic effects of the combination treatment of Bac-ELP1-H1 and TMZ on U87 cells. (**A**) Comparison of cell viability of U87 cells after treatment with the combination of Bac-ELP1-H1 and TMZ to treatment with each compound individually. (**B**,**C**) Three-dimensional Loewe and HSA synergy plots of the combination treatment of Bac-ELP1-H1 and TMZ on U87 cells. Red indicates synergy, while green indicates antagonism for the combination treatment.

**Figure 2 bioengineering-10-00718-f002:**
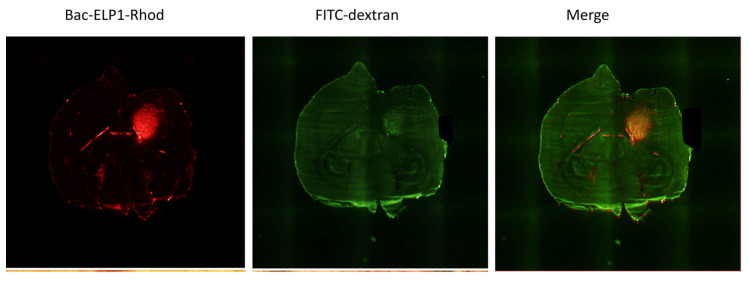
Localization of Bac-ELP-H1 in brain tumor. Fluorescence microscopy images of brain sections from mice injected with rhodamine-labeled Bac-ELP-H1 and FITC-dextran to mark the perfused vasculature. The (**left**) panel shows the distribution of rhodamine-labeled polypeptides within the tumor. The (**middle**) panel shows the perfused vasculature marked by infusion of high molecular weight dextran. The (**right**) panel is the merged image of the tumor and perfused vasculature. Bac-ELP-H1 polypeptides were detected in the tumor relative to autofluorescence, as evidenced by bright staining of the tumors.

**Figure 3 bioengineering-10-00718-f003:**
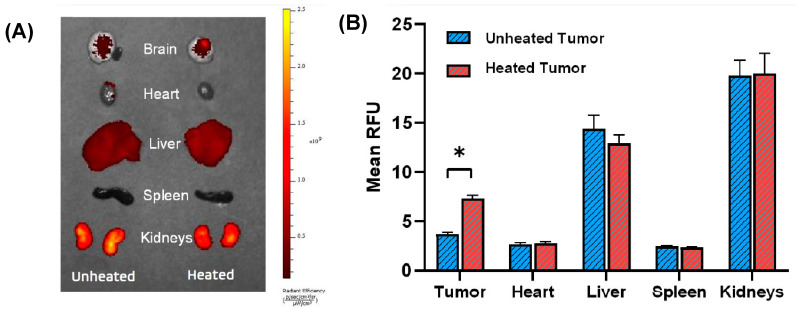
Enhancement of Bac-ELP-H1 tumor uptake by thermal targeting. Ex vivo whole organ fluorescence imaging was used to determine tumor and organ levels following IV administration of rhodamine-labeled Bac-ELP-H1 with or without hyperthermia. To confirm the effects of thermal targeting in mouse glioblastoma models, tumors were heated using the thermal cycling protocol, and tumor deposition was examined 4 h after the injection using an IVIS spectrum animal imager. (**A**) Representative images of ex vivo fluorescence of major organs collected from tumor-bearing mice. The heat map scale was set as radiant efficiency. The scale bars indicate the level of fluorescence intensity. (**B**) Average radiant efficiency calculated from ex vivo fluorescence after selecting appropriate regions of interest (ROIs) using IVIS software. Quantification of fluorescence from tumors and all major organs is expressed in relative fluorescence units (RFU). The data are presented as the mean ± standard error of the mean (s.e.m.). Statistical analysis showed that fluorescence levels were significantly different between the unheated and heated tumors *(*p* < 0.01, Student’s *t*-test, *n* = 7–9 mice/group).

**Figure 4 bioengineering-10-00718-f004:**
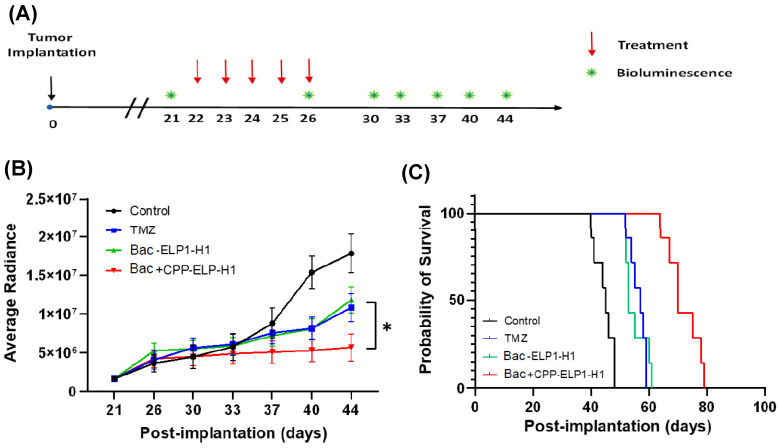
Bac-ELP-H1 and TMZ combination therapy reduces tumor burden in vivo. (**A**) Timeline illustrating the treatment schedule of mice implanted with U87-MG luciferase transfected cells (2 × 104 in 2 μL of PBS) and intracranially injected into 4–6 week-old mice using a stereotactic frame. After 20 days, mice were randomized into four groups (n = 7 per group) and treated with vehicle, TMZ (32 mg/kg BW) by oral gavage, Bac-ELP-H1(30 mg/kg BW) by IV, or a combination of the two therapies for five consecutive days. (**B**) The tumor volume (average radiance) was measured using IVIS imaging, and data represent the mean total flux (±SD). A paired *t*-test was used to assess the significant difference between means; * *p* < 0.05. (**C**) The survival of mice is shown as Kaplan–Meier survival curves. The log-rank test was used to assess statistical significance (n = 7); *p* < 0.05. TMZ or Bac-ELP-H1 treatment alone was compared to the combination therapy.

## Data Availability

The data presented in this study are available in the article and on request from the corresponding author.

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
