# Peer review of "Targeted c-Myc Inhibition and Systemic Temozolomide Therapy Extend Survival in Glioblastoma Xenografts"

_bioengineering, 2023, doi:10.3390/bioengineering10060718_

Round 1

Reviewer 1 Report

The article submitted for expert review is a study following previous work of the authors, on the treatment of glioblastoma. The authors use the therapeutic peptide (Bac-ELP1-H1), designed in previous work, extending to the U87 glioblastoma cell line and study the efficacy of Bac-ELP1-H1/hyperthermia treatment, as well as the combination treatment temozolomide - Bac-ELP1-H1, to suppress tumor growth and prolong survival. in athymic mice. The study shows that the combined treatment of Bac-ELP1-H1 - temozolomide acted synergistically to improve survival in mice and showed better efficacy in reducing tumor progression than the components taken alone. The study described also demonstrates the effectiveness of hyperthermia in facilitating the accumulation of Bac-ELP1-H1 protein at the tumor site. The numerous and well-presented results suggest that the combination of targeted c-myc inhibitor biopolymer with systemic treatment with temozolomide may be a promising alternative treatment.

The study is perfectly described, the numerous results allow the authors to develop a perfect discussion. At the level of the presentation of the results, some figures must be improved for readability.

The authors indicate that current treatments cause side effects but what about their treatment proposal. Do these new protocols avoid any side effects already known or could generate new side effects. The potential treatment to propose; Is it without side effects. No impact on BBB?
Another question that should be addressed at least at the end of the discussion; is this easily transposable to the human body in vivo; no formulation/galenic constraints?

Reviewer 3 Report

Dhungel et al. reported the use of a thermal-responsive therapeutic peptide and TMZ to synergistically treat glioblastoma. The peptide localized at tumor site and inhibited tumor growth. As a result, the reported therapy extended the median survival of tumor-bearing mice. The work fits the scope of Bioengineering, however, the following questions should be addressed prior to accepting this article.

1.     Figure 1: Please include the hyperthermia experiment, and increase the resolution of all figures.

2.     Figure 3: Please label the ex vivo result with “heated” and “unheated”.

3.     Figure 4: What is CPP? It is not referred anywhere in the manuscript. Is there any chance to check the synergistic effect on living animals? The authors emphasized the side effects of using TMZ alone but without showing any results about avoiding the conventional side effects by the reported strategy. I would highly suggest adding this experiment.

4.     Please double-check the method part (line 137): are 500 cells the correct number for the experiment?

1.    Typos: line 38 ”(;”, line 267 “Brain Tumor”, line 283 “Figure 1”.

2.     Reference format: line 81-82 “Louis et al., 2007; Scherer et al., 2019”.

Round 2

Reviewer 3 Report

The manuscript looks pretty good now.